# Quantifying absolute gene expression profiles reveals distinct regulation of central carbon metabolism genes in yeast

Rosemary Yu[1,2], Egor Vorontsov[3], Carina Sihlbom[3], Jens Nielsen[1,2,4,5]*

[1]Department of Biology and Biological Engineering, Chalmers University of Technology, Gothenburg, Sweden; [2]Novo Nordisk Foundation Center for Biosustainability, Chalmers University of Technology, Gothenburg, Sweden; [3]Proteomics Core Facility, Sahlgrenska Academy, University of Gothenburg, Gothenburg, Sweden; [4]Novo Nordisk Foundation Center for Biosustainability, Technical University of Denmark, Lyngby, Denmark; [5]BioInnovation Institute, Copenhagen, Denmark

**Abstract** In addition to controlled expression of genes by specific regulatory circuits, the abundance of proteins and transcripts can also be influenced by physiological states of the cell such as growth rate and metabolism. Here we examine the control of gene expression by growth rate and metabolism, by analyzing a multi-omics dataset consisting of absolute-quantitative abundances of the transcriptome, proteome, and amino acids in 22 steady-state yeast cultures. We find that transcription and translation are coordinately controlled by the cell growth rate via RNA polymerase II and ribosome abundance, but they are independently controlled by nitrogen metabolism via amino acid and nucleotide availabilities. Genes in central carbon metabolism, however, are distinctly regulated and do not respond to the cell growth rate or nitrogen metabolism as all other genes. Understanding these effects allows the confounding factors of growth rate and metabolism to be accounted for in gene expression profiling studies.

*For correspondence:
nielsenj@chalmers.se

## Introduction

In the presence of environmental or genetic perturbations, differential expression of genes, orchestrated by dedicated regulatory circuits, shapes the physiological responses of the cell. Common physiological responses to perturbations, for example in response to stress or during oncogenic transformation, often include changes in the cell growth rate and metabolism. Both growth rate and metabolism in turn influence gene expression: a faster growing cell doubles its transcriptome and proteome in a shorter amount of time, and synthesis of transcripts and proteins demand energy and building blocks to be supplied through metabolism. Seminal studies in the field (*Brauer et al., 2008*; *Boer et al., 2010*; *Slavov and Botstein, 2011*) have previously examined the interaction between growth rate, metabolism, and gene expression, using microarrays and relative-quantitative metabolomics, in the eukaryal model organism *Saccharomyces cerevisiae*. However, absolute-quantitative concentrations of transcripts, proteins, and metabolites are needed to understand the precise influence of growth rate and metabolism on gene expression and to examine the molecular mechanisms underlying these effects. Understanding these effects would allow studies of specific environmental or genetic perturbations to uncouple the perturbation-specific regulation of gene expression from gene expression control exerted by differences in cell growth or metabolism. In bioengineering applications, this would also allow synthetic gene circuits with complex behaviors to be designed, a key goal of synthetic biology (*Shahrezaei and Marguerat, 2015*; *Slavov et al., 2011*).

Herein we examine the absolute-quantitative transcriptomic, proteomic, and intracellular amino acid concentrations, in a total of 22 steady-state yeast chemostat cultures in biological triplicates, designed to orthogonally probe the effects of growth rate and nitrogen metabolism on gene expression. The 22 chemostat conditions consist of 14 which we have performed in this study and 9 which were performed by *Xia, 2019a*; *Xia, 2019b*, with one condition being a duplicate between the two studies (*Figure 1A*). We found that growth rate and metabolism influence the expression of ~90% of genes; however, genes in central carbon metabolism represented a major group of genes that were exceptions to these effects. The gene expression changes associated with growth rate were highly coordinated between transcript abundance and protein abundance and reflected the availabilities of the transcription and translation machineries. In contrast, the coordination between transcript and protein concentrations was lost when amino acid and nucleotide availabilities were modulated experimentally without changing the cell growth rate. Finally, by re-analyzing gene expression

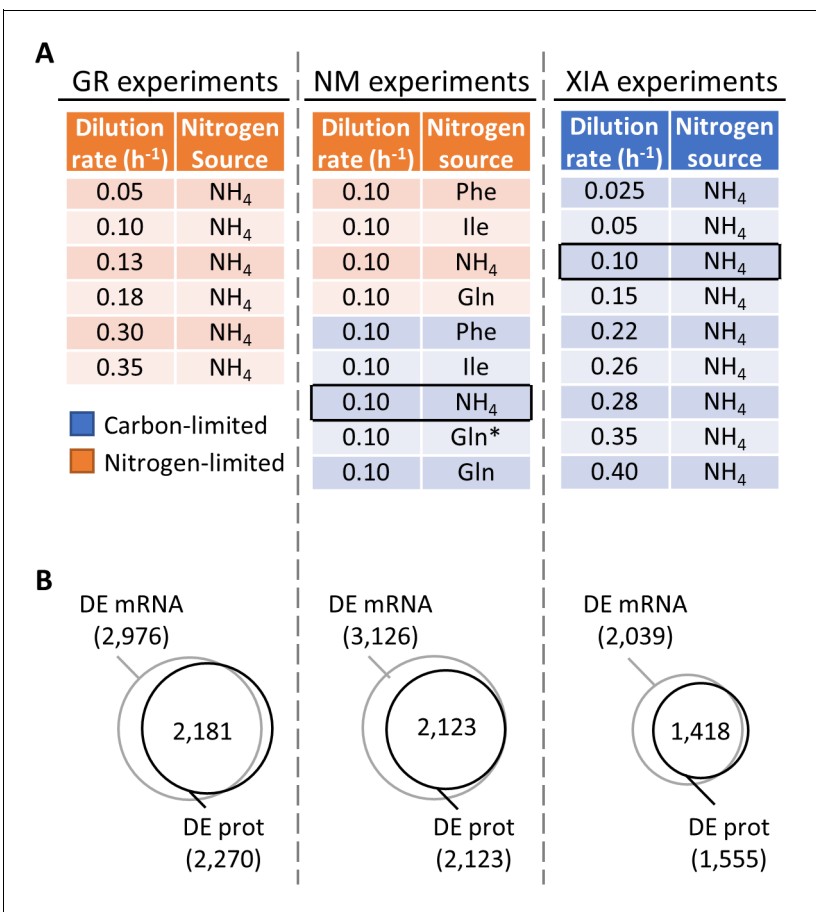

**Figure 1.** Regulation of gene expression by the physiological state of the cell. (**A**) Experimental design to orthogonally probe the effects of growth rate and nitrogen metabolism on gene expression. Cells were grown in chemostats at controlled growth rates and media composition. GR, growth rate; NM, nitrogen metabolism; (*Xia, 2019a*; *Xia, 2019b*). In NM experiments, carbon-limited conditions (blue rows), the 'Gln' condition, and the 'Gln*' condition differ in the concentration of Gln and glucose in the chemostat feed media; see *Supplementary file 1a* for full details. Rows in black boxes represent a duplicated condition between the chemostats performed in this study and in *Xia, 2019a*; *Xia, 2019b*. (**B**) Number of differentially expressed (DE; FDR < 0.01 by one-way ANOVA) genes at mRNA and protein (prot) levels in the GR experiments, NM experiments, and XIA experiments, showing that a large number of genes are regulated by growth rate and nitrogen metabolism.

The online version of this article includes the following figure supplement(s) for figure 1:

**Figure supplement 1.** Total and sample-wise protein–mRNA correlations.

**Figure supplement 2.** Measured absolute quantity of proteins and transcripts for subunits of protein complexes.

profiles of a distantly related yeast, *Schizosaccharomyces pombe*, and of the human Burkitt's lymphoma cell line P493–6, we demonstrated that these effects on gene expression by the cell growth rate and metabolism can be uncoupled from gene expression regulation by specific environmental and genetic perturbations, allowing novel biological insights in gene expression regulation to be uncovered.

## Results

### Data description

We performed a series of 14 steady-state chemostat cultures of *S. cerevisiae*, which orthogonally modulated either the growth rate (GR experiments) or the nitrogen metabolism (NM experiments) of yeast (*Figure 1A*, *Supplementary file 1a*). Nitrogen metabolism was of particular interest since nitrogen is a critical component of amino acids and nucleotides. The nitrogen sources in the NM experiments were chosen to represent those that are preferred ($NH_4$ and Glu) and non-preferred (Phe and Ile), as previously defined (*Godard et al., 2007*). Depending on the growth condition, the total protein content in the dry cell weight ranged between 20.6% and 59.4%, and the total RNA content ranged between 1.8% and 8.9%, (*Supplementary file 1a*), in agreement with previous observations of total protein and RNA content with changing growth rate and nitrogen source (*Verduyn, 1991*; *Larsson et al., 1993*; *Yu et al., 2020a*). We then profiled the absolute-quantitative transcriptomic and proteomic abundances (mmol/gDW) of these chemostat cultures in biological triplicates, generating a multi-omics dataset containing 3127 transcript–protein pairs across 14 conditions (*Supplementary file 1b*). In both GR and NM experiments, we found >2,000 genes to be differentially expressed at both the transcript and protein levels with false discovery rate (FDR) < 0.01 (one-way ANOVA comparing all conditions; *Figure 1B*), indicating that gene expression is subject to extensive control by both growth rate and nitrogen metabolism. A total of 1493 genes (48%) in the GR experiments, and 1959 genes (63%) in the NM experiments, were differentially expressed not only significantly, but by more than twofold. In *Figure 1—figure supplement 1*, we present the total protein–RNA correlation of these chemostat cultures, with a Pearson r of 0.51 and a Spearman ρ of 0.27, and sample-wise proteome–transcriptome correlations, with a median Pearson r of 0.40 and Spearman ρ of 0.63, agreeing well with previous studies (*Yu et al., 2020a*; *Lahtvee et al., 2017*; *Schwanhäusser et al., 2011*; *Marguerat et al., 2012*). In *Figure 1—figure supplement 2*, we present the measured absolute quantities of subunits for several complexes (*Yu et al., 2020a*; *Taggart and Li, 2018*), showing that our measured protein abundances agree with the known subunit stoichiometry within 1–2 orders of magnitude, typical of iBAQ-based proteomics quantitation (*Yu et al., 2020a*; *Schwanhäusser et al., 2011*; *Björkeroth et al., 2020*). This variability in the proteomics data reflects a mixed effect of inaccuracies in the proteomics methodology, and proteins that are partially synthesized, partially degraded, or not being part of its designated protein complex. For f1f0 ATP synthase, ATP15 and TIM11 were 4 orders of magnitude lower than the abundance of other subunits (*Figure 1—figure supplement 2*), likely reflecting the difficulty in extracting and quantifying membrane-embedded proteins.

We further mined absolute-quantitative transcriptomics and proteomics data from *Xia, 2019a*; *Xia, 2019b* (XIA experiments), wherein *S. cerevisiae* were grown at nine different dilution rates with glucose being the limiting nutrient (*Figure 1A,B*). Of these, one condition (carbon-limited, nitrogen source $NH_4$, dilution rate 0.10 $h^{-1}$) was duplicated in the chemostats that we performed (*Figure 1A*); thus, the number of unique conditions examined in this study is a total of 22. Data from the XIA experiments, containing paired transcript-protein abundance of 2235 genes (*Supplementary file 1c*), allowed us to validate our findings in the GR experiments in a different nutrient limitation setting.

Leveraging these absolute-quantitative data, we performed a series of exploratory analyses with the goal of describing the control of (1) transcription and (2) translation with respect to growth rate; and (3) transcription and (4) translation with respect to nitrogen metabolism. As transcript and protein abundances span several orders of magnitude, for ease of visualization we present most of our analyses using scaled expression levels; details of the scaling done can be found in the Materials and methods section, and the absolute concentrations can be found in supplementary files as indicated. We also suggest mechanisms and consequences of these observations, by examining

the abundance of RNA polymerase II, ribosomes, and intracellular amino acid levels. The lessons learned from these analyses are collected into a conceptual model. We note however that this does not imply direct causation, rather hypotheses which should be tested by further experiments. Finally, we showcase the significance of understanding the effect of growth rate and metabolism on gene expression, by accounting for these confounding factors in a re-analysis of two previously published gene expression profiling datasets.

## Growth rate regulates a large number of transcripts with exceptions in central carbon metabolism

We first considered the regulation of mRNA abundance by growth rate. We used 25 clustering indices (*Charrad et al., 2014*) to determine the best clustering scheme for transcript abundance in the GR experiments and found that they fell into an optimal number of three clusters (*Figure 2A*). The list of genes in each of the clusters can be found in *Supplementary file 1b* and filtering by the column 'GR.cluster'. Expression of transcripts in GR.cluster.1 and GR.cluster.3, together including 2895 genes (92%), exhibited strong associations with growth rate (*Figure 2B*). In contrast, expression of 232 transcripts (7%) in GR.cluster.2 did not increase with increasing growth rate (*Figure 2C*). These growth rate-independent genes in GR.cluster.2 were enriched in GO-slim terms (*Cherry et al.,*

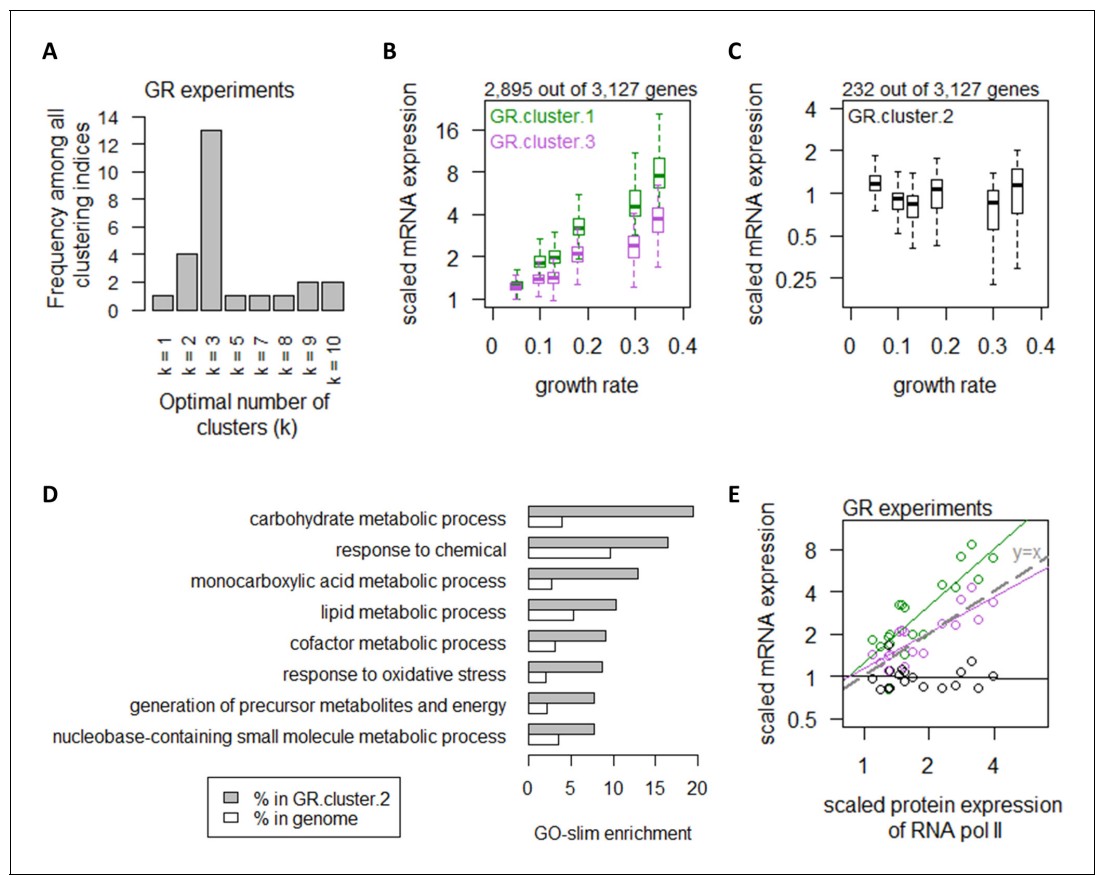

**Figure 2.** Growth rate regulates a large number of transcripts with exceptions in central carbon metabolism. (**A**) Using 25 clustering indices, we found that most indices suggest an optimal number of three clusters for transcript abundance in the GR experiments. (**B**) Abundance of transcripts in GR. cluster.1 (green) and GR.cluster.3 (purple). Center line, median; box limits, upper and lower quartiles; whiskers, 1.5× interquartile range. (**C**) Abundance of transcripts in GR.cluster.2. Center line, median; box limits, upper and lower quartiles; whiskers, 1.5× interquartile range. (**D**) GO-slim enrichment of genes in GR.cluster.2 showing enrichment in GO-slim terms related to CCM, among others. (**E**) Expression of RNA polymerase II protein abundance and mRNA abundance of the three clusters in the GR experiments. Colors are as (**B**) and (**C**). Median mRNA expression values in each cluster and median protein expression of RNA polymerase II are shown. Gray dashed line represents y=x.

The online version of this article includes the following figure supplement(s) for figure 2:

**Figure supplement 1.** Global control of transcript abundance by growth rate.

*2012*) related to central carbon metabolism (CCM), including 'carbohydrate metabolic process' and 'generation of precursor metabolites and energy' (*Figure 2D*). We further observed that genes in GR.cluster.1 and GR.cluster.3 closely followed the protein abundance of RNA polymerase II (*Carlberg and Molnár, 2014*; *Figure 2E*), suggesting that transcript abundance of these 92% of genes reflected the availability of RNA polymerase II, consistent with previous studies (*Klumpp and Hwa, 2008*; *Klumpp et al., 2009*; *Heldt et al., 2018*). In contrast, the 232 genes in GR.cluster.2 exhibited slightly decreased abundance with increasing RNA polymerase II expression (*Figure 2E*), indicating that these genes were regulated by a mechanism independent from growth rate-associated changes in RNA polymerase II.

To confirm these findings, we performed similar analyses using data from the XIA experiments (*Xia, 2019a*; *Xia, 2019b*), where the optimal number of clusters was calculated to be two clusters (*Figure 2—figure supplement 1A*). The list of genes in each of the clusters in the XIA experiment can be found in *Supplementary file 1c* and filtering by the column 'XIA.cluster'. XIA.cluster.2 contained 1964 genes (88%), which exhibited growth rate-dependent transcript abundance (*Figure 2—figure supplement 1B*), while XIA.cluster.1 contained 271 genes (12%) showing growth rate-independence. Genes in XIA.cluster.1 were enriched in CCM GO-slim terms (*Figure 2—figure supplement 1C,D*), including 'carbohydrate metabolic process' and 'generation of precursor metabolites and energy', validating our results in the GR experiments. Of note, GO-slim terms related to CCM ('carbohydrate metabolic process'; 'cellular respiration'; and 'generation of precursor metabolites and energy') were enriched among the most abundant 10% genes in both datasets (*Figure 2—figure supplement 1E–F*), demonstrating that the transcript expression of vast majority of genes were controlled by the cell growth rate, but a distinct mechanism regulated a small number of CCM-related transcripts that are highly abundant.

## Growth rate regulates protein abundance in coordination with transcript abundance

We next examined the relationship between mRNA and protein abundance of each gene in the GR experiments (*Figure 3*). As the underlying distribution of the mRNA and protein abundances in these analyses were non-normal ($p_{Shapiro-Wilk} < 0.01$) for a large portion of genes, we used Spearman correlations in the following analyses. We note that Spearman correlation coefficients ($\rho$) were closely related to Pearson correlation coefficients (r) (*Figure 3—figure supplement 1*), and as such our

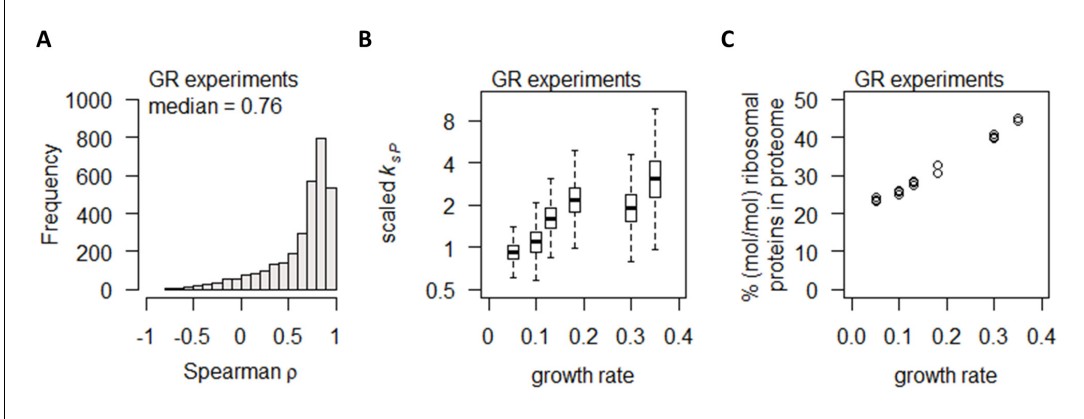

**Figure 3.** Growth rate regulates protein abundance in coordination with transcript abundance. (A) Spearman correlation of absolute mRNA and protein abundances for each gene is calculated in the GR experiments, and the distribution is shown demonstrating overall high correlation. (B) Protein translation rate ($k_{sP}$) in the GR experiments increases with growth rate by about fourfold. Center line, median; box limits, upper and lower quartiles; whiskers, 1.5× interquartile range. (C) Relative ribosomal protein abundance in the GR experiments was calculated as the sum of all detected ribosomal proteins and normalized to the total protein content, showing a linear increase with increasing growth rate.

The online version of this article includes the following figure supplement(s) for figure 3:

**Figure supplement 1.** Correlation between protein and mRNA abundance for each gene showing that Spearman and Pearson correlations track well with each other.

observations can be easily compared to previous studies that have used Pearson correlations. In *Supplementary file 1b*, we report both Spearman ρ and Pearson r values.

Several previous studies have suggested that, comparing the mRNA and protein abundances of the same gene across different experimental conditions, generally there is a high correlation between protein and mRNA abundance (*Lahtvee et al., 2017*; *Yu and Nielsen, 2019*; *Newman et al., 2006*; *Csárdi et al., 2015*). Here, in the GR experiments, we observed high protein–mRNA correlations consistent with these studies, with a median ρ of 0.76 (*Figure 3A*) and 2379 genes (78%) having ρ > 0.5. We then calculated the protein translation rate for each gene ($k_{sP}$, in the unit of protein/mRNA/h) (*Lahtvee et al., 2017*; *Schwanhäusser et al., 2011*) by:

$$k_{sP,j} = \frac{C_{prot,j} \cdot \left(k_{dP,j} + \mu\right)}{C_{mRNA,j}}$$

where $k_{sP,j}$ is the protein translation rate for gene $j$; $C_{prot,j}$ is the concentration of protein $j$; $k_{dP,j}$ is the protein degradation rate for gene $j$, measured by *Lahtvee et al., 2017*; $\mu$ is the specific growth rate; and $C_{mRNA,j}$ is the concentration of mRNA $j$. For those genes where $k_{dP,j}$ was not determined in *Lahtvee et al., 2017*, the median of all $k_{dP}$ in that dataset was used. We found that overall $k_{sP}$ is increased with increasing growth rate (*Figure 3B*). Concurrently, we observed an increase in the % (mol/mol) of ribosomal proteins in the total proteome (*Figure 3C*), consistent with previous reports (*Yu and Nielsen, 2019*; *Metzl-Raz et al., 2017*) and suggesting that the increase in $k_{sP}$ in the GR experiments were linked to the availability of ribosomes. Taken together, our data indicate that increasing growth rate modulated gene expression via coordinated increases in both transcription and translation, by increasing the availability of RNA polymerase II and ribosomes.

## Nitrogen metabolism regulates a large number of transcripts with exceptions in central carbon metabolism

In the NM experiments, using the same 25 clustering indices (*Charrad et al., 2014*), we calculated that transcript expression fell into an optimal number of two clusters (*Figure 4A*). The list of genes in each of the clusters in the NM experiments can be found in *Supplementary file 1b* and filtering by the column 'NM.cluster'. In NM.cluster.1, a total of 3011 genes (96%) exhibited increasing expression levels when cells were grown on preferred nitrogen sources ($NH_4$ and Gln) compared to non-preferred nitrogen sources (Phe and Ile), under carbon-limiting conditions (*Figure 4B*). In contrast, 116 transcripts (4%) in NM.cluster.2 did not show the same expression pattern (*Figure 4C*). These genes were enriched in processes related to CCM, as well as amino acid metabolic processes (*Figure 4D*).

Examining RNA polymerase II levels, we found that changes in transcript abundance was not associated with RNA polymerase II availability in the NM experiments (*Figure 4E*). As the increased transcripts in the majority of genes when cells were grown on preferred nitrogen sources (*Figure 4B*) must be supported by increased biosynthesis of nucleotides, we therefore examined the intracellular concentrations of Ser and Gly, which are substrates for purine synthesis (*Figure 5A*), as well as Gln and Asp, which are upstream of both purine (*Figure 5A*) and pyrimidine synthesis (*Figure 5B*). We found that transcript abundance in the NM experiments tracked closely with intracellular Ser and Gly concentrations (*Figure 5C*), but not with Gln or Asp (*Figure 5D*). Moreover, in absolute-quantitative terms (*Supplementary file 1d*), Gln and Asp were present at much higher intracellular concentrations (mean of 166.7 and 12.4 μmol/gDW, respectively, across all NM experiments) compared to Gly and Ser (mean of 1.5 and 2.8 μmol/gDW, respectively, across all NM experiments). Together, this suggests that intracellular Gly and Ser were likely limiting for purine synthesis, which in turn constrained transcript abundance when the nitrogen source was scarce or non-preferred.

## Nitrogen metabolism regulation of protein abundance is not coordinated with transcript abundance or ribosome abundance

In the NM experiments, the gene-specific protein–mRNA correlations were surprisingly poor, with a median ρ of 0.07 (*Figure 6A*) and most genes with −0.5 < ρ < 0.5 (2763 genes, 88%). To check whether the poor correlations stem from lower range of differential gene expression, we split the total of 3127 genes into quartiles by mRNA differential expression (fold-change of largest to smallest

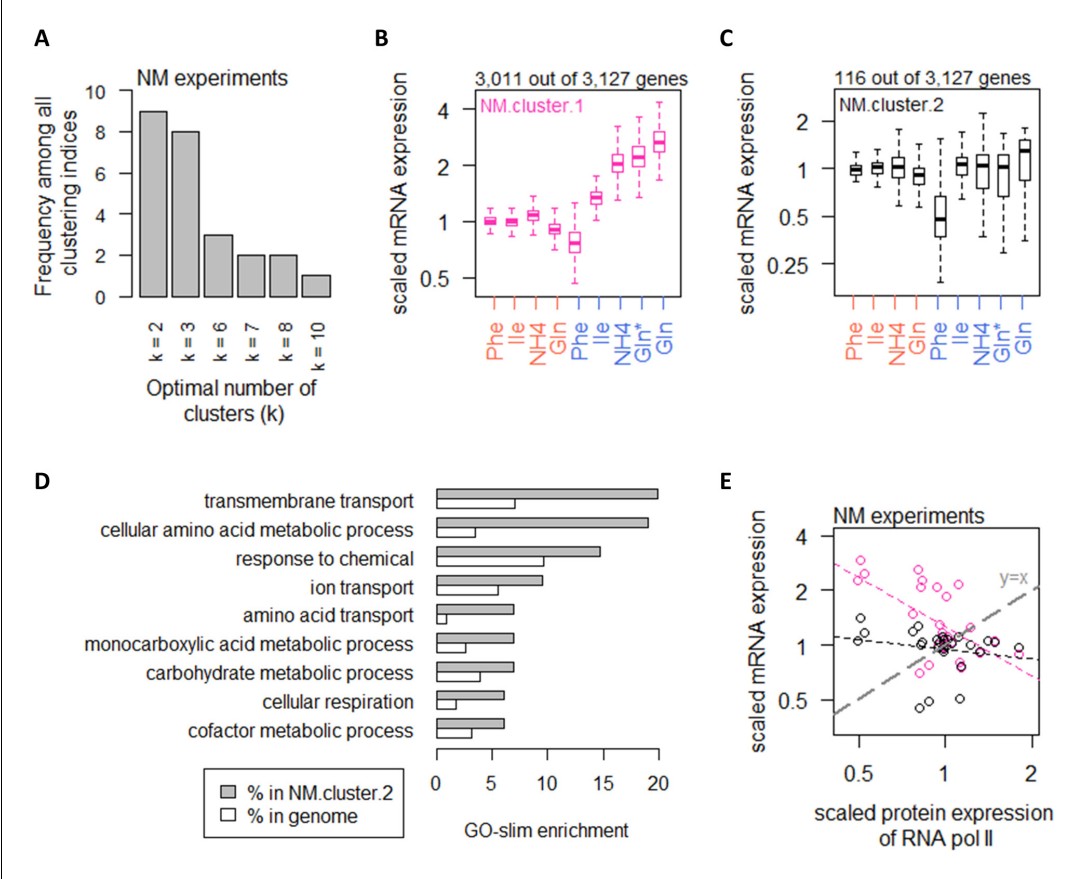

**Figure 4.** Nitrogen metabolism regulates a large number of transcripts with exceptions in CCM and amino acid metabolic processes. (**A**) Using 25 clustering indices, we found that most indices suggest an optimal number of two clusters for transcript abundance in the NM experiments. (**B**) Abundance of transcripts in NM.cluster.1. Center line, median; box limits, upper and lower quartiles; whiskers, 1.5× interquartile range. In NM experiments, carbon-limited conditions (blue rows), the 'Gln' condition, and the 'Gln*' condition differ in the concentration of Gln and glucose in the chemostat feed media; see *Supplementary file 1a* for full details. (**C**) Abundance of transcripts in NM.cluster.2. Center line, median; box limits, upper and lower quartiles; whiskers, 1.5× interquartile range. In NM experiments, carbon-limited conditions (blue rows), the 'Gln' condition, and the 'Gln*' condition differ in the concentration of Gln and glucose in the chemostat feed media; see *Supplementary file 1a* for full details. (**D**) GO-slim enrichment of genes in NM.cluster.2 showing enrichment in GO-slim terms related to CCM, among others. (**E**) Expression of RNA polymerase II protein abundance and mRNA abundance of the two clusters in the NM experiments. Colors are as (**B**) and (**C**). Median mRNA expression values in each cluster and median protein expression of RNA polymerase II are shown. Gray dashed line represents y=x.

measured value) and found that, even in the fourth quartile where the range of differential expression was >5.3-fold, the protein–mRNA correlations remained very low, with a median ρ of 0.24 (*Figure 6—figure supplement 1A*). Similar results were found when genes were split into quartiles by protein differential expression (*Figure 6—figure supplement 1B*), indicating that the poor protein–mRNA correlations observed for most genes in the NM experiments cannot be explained by the range of differential gene expression. Calculating the protein translation rate $k_{sP}$, we found that gene-specific $k_{sP}$ in the NM experiments were generally reduced when cells were grown on preferred nitrogen sources, in both nitrogen-limited cultures and carbon-limited cultures (*Figure 6B*). These changes in $k_{sP}$ occurred without any changes in the % of ribosomal proteins in the total proteome (*Figure 6C*). As the growth rate of the NM experiments was controlled to a constant 0.1 h$^{-1}$ (*Figure 1A*), this indicates that the % of ribosomal proteins in the total proteome was a direct reflection of the cell growth rate, while nitrogen metabolism modulated protein translation without modifying the abundance of ribosomes.

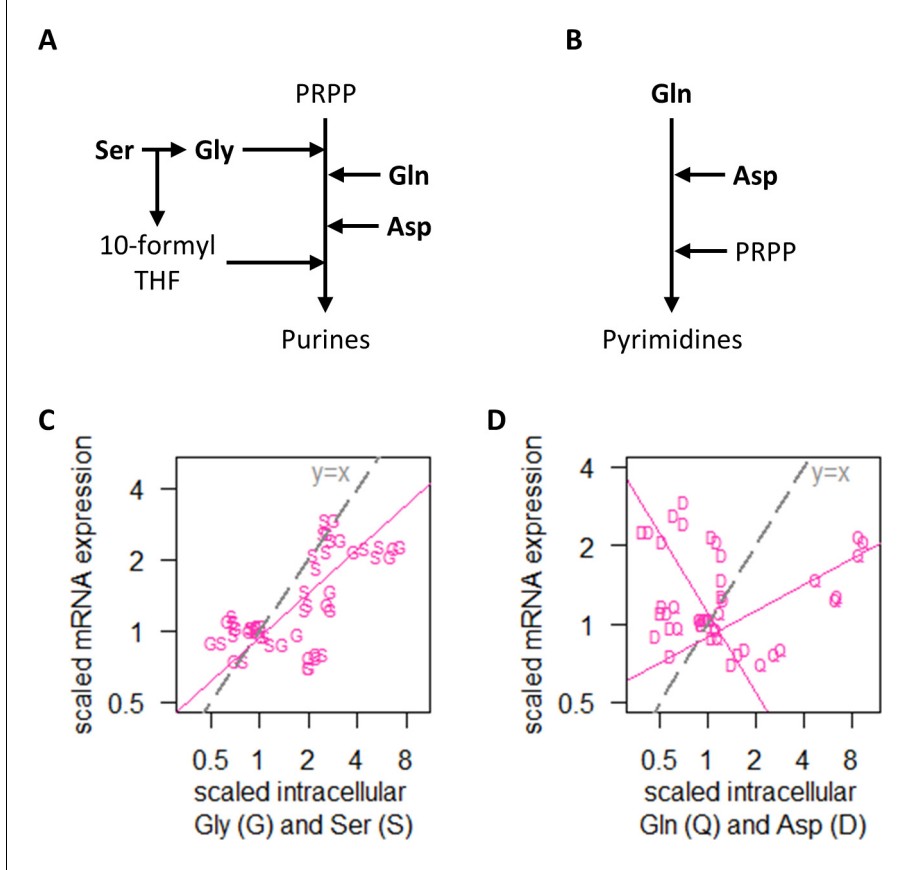

**Figure 5.** Regulation of transcript abundance by nitrogen metabolism hinges on Ser and Gly intracellular concentrations. (**A**) Simplified pathway of purine synthesis showing dependencies on Ser, Gly, Gln, and Asp. (**B**) Simplified pathway of pyrimidine synthesis showing dependencies on Gln and Asp only. (**C**) Intracellular concentrations of Gly (G) and Ser (S), and RNA abundance of genes in NM.cluster.1, showing that transcript abundance in the NM experiments track closely with intracellular Ser and Gly concentrations. Median mRNA expression values of NM.cluster.1 are shown. (**D**) Intracellular concentrations of Gln (Q) and Asp (D), and RNA abundance of genes in NM.cluster.1, showing that transcript abundance for most genes in the NM experiments do not track with intracellular Gln and Asp concentrations. Median mRNA expression values of NM.cluster.1 are shown.

## The correlation between protein and transcript abundance is especially poor for central carbon metabolism genes

Combining all gene expression measurements in both GR and NM experiments, we found that the protein–mRNA correlations remained poor, with a median ρ of 0.22 (*Figure 7A*). This suggests that, while the most genes were differentially expressed in response to growth rate and/or metabolism at the transcript level, this does not necessarily translate into concordant changes in protein abundance. We also considered the possibility that relative abundance of proteins and transcripts may be well correlated, while their absolute abundance may be irrelevant. However, the correlation of mRNA relative-abundance (% mol/mol of total mRNA) with protein relative-abundance (% mol/mol of total protein) was also poor, with a median ρ of 0.34 (*Figure 7—figure supplement 1*). Thus, transcription and translation appear to be two distinct points of gene expression regulation that overall are not concordantly modulated.

The large range of ρ spanning from around −0.5 to 1 (*Figure 7A*) nevertheless indicates that for some genes, the correlation between protein and transcript levels can be very high, while for others, this correlation would be especially poor. To examine these relationships, we tested for enrichment of GO-slim terms in 200-gene sliding windows of increasing ρ (*Yu et al., 2020a*; *Marguerat et al., 2012*). We found that genes with high correlations between protein and mRNA abundance (median

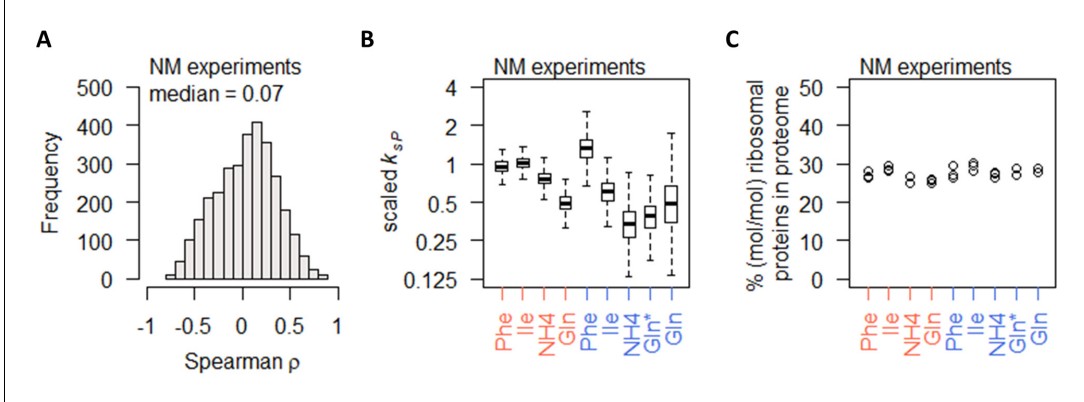

**Figure 6.** Nitrogen metabolism regulation of protein abundance is not coordinated with transcript abundance or ribosome abundance. (A) Spearman correlation of absolute mRNA and protein abundances for each gene is calculated in the NM experiments, and the distribution is shown demonstrating overall poor correlation. (B) Protein translation rate ($k_{sP}$) in the NM experiments decreases when cells were grown on preferred nitrogen sources ($NH_4$ and Gln) compared to non-preferred nitrogen sources (Phe and Ile). In nitrogen-limited cultures (orange), there is an overall 2-fold decrease, and in carbon-limited cultures (blue), there is an overall 4-fold decrease. Center line, median; box limits, upper and lower quartiles; whiskers, 1.5× interquartile range. In carbon-limited conditions (blue), the 'Gln' condition and the 'Gln*' condition differ in the concentration of Gln and glucose in the chemostat feed media; see **Supplementary file 1a** for full details. (C) Relative ribosomal protein abundance (sum of all detected ribosomal proteins normalized to total protein) is constant in the NM experiments, where the growth rate is controlled to a constant as seen in **Figure 1A**. In carbon-limited conditions (blue), the 'Gln' condition and the 'Gln*' condition differ in the concentration of Gln and glucose in the chemostat feed media; see **Supplementary file 1a** for full details.

The online version of this article includes the following figure supplement(s) for figure 6:

**Figure supplement 1.** Comparing protein–mRNA correlations in the NM experiments with the range of differential expression.

ρ of 0.4–0.7 in 200-gene windows) are enriched in GO-slim terms related to amino acid metabolic processes, and processes related to protein translation, including ribosome biogenesis; rRNA processing; tRNA aminoacylation; and several others (**Figure 7B**). On the other hand, genes with poor protein–mRNA correlations (median ρ of −0.2 to 0.2 in 200-gene windows) were enriched in GO-slim terms related to CCM ('carbohydrate metabolic process', 'cellular respiration', and 'generation of precursor metabolites and energy'; **Figure 7B**), indicating distinct roles of transcription and translation in regulating CCM gene expression, which reflects the unique and complex mechanisms that regulate this important part of metabolism (**Yu et al., 2020a**).

We then tested whether genes with different protein–transcript correlations have different proportions of specific amino acids in their protein sequences. Splitting genes into 200-gene brackets of increasing ρ, we found that protein–transcript correlations vary with the sequence proportion of several amino acids (**Figure 7C**). Of note is that genes with low protein–transcript correlations contain higher proportions of Gln, Glu, Asn, and Asp (**Figure 7C**), which are central to the metabolism of all other amino acids as well as nucleic acids. Thus, it is possible that the uncoupling of translational regulation from transcriptional regulation in these genes represents a mechanism to allow intracellular amino acids to modulate protein abundance. We also found that the codon usage of genes with low protein–transcript correlation tend to have higher usage of purines (**Figure 7D**), suggesting that nucleic acid levels may also play a role in controlling the transcript abundance of these grenes and contributing to the uncoupling between transcription and translation.

## Model of gene expression control by growth rate and metabolism

In **Figure 8**, we collect the lessons learned from all of the above analyses into a conceptual model. Information flow in the central dogma and a simplified flow of nitrogenous metabolites are given as reference points. Analysis of data in the GR experiment revealed that changes in the abundance of transcripts and proteins with respect to the cell growth rate are highly correlated, and both are closely associated with levels of RNA polymerase II and ribosomes in the cell (**Figure 8**, green text and arrows). This general rule however does not apply to a small group of genes enriched in CCM (**Figure 8**, note *a*). In contrast, nitrogen metabolism influences gene expression via a mechanism that is independent of RNA polymerase II and ribosome levels, with transcript and protein

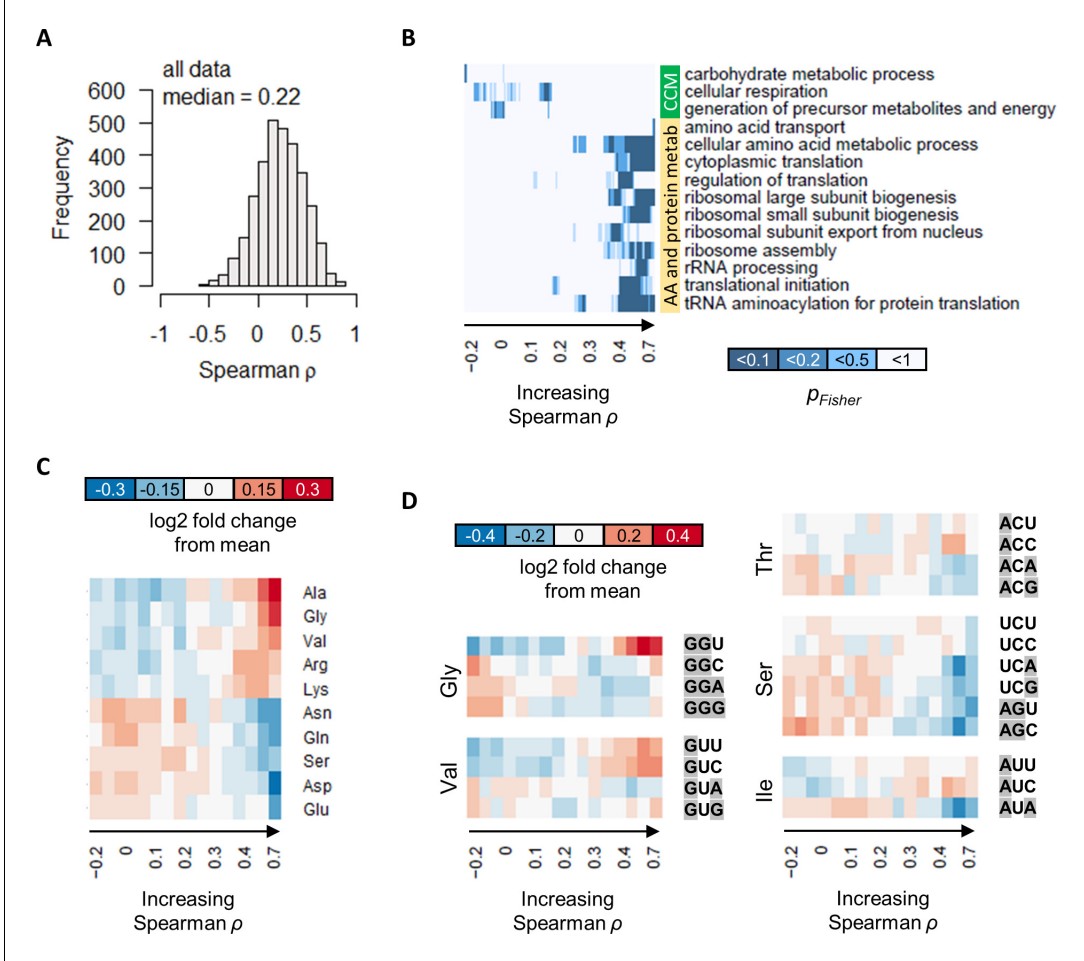

**Figure 7.** The correlation between protein and transcript abundance is especially poor for central carbon metabolism genes. (A) Spearman correlation of absolute-quantitative mRNA and protein abundances for each gene was calculated using data from both the GR and the NM experiments combined, and the distribution is shown demonstrating overall poor correlation. (B) Enrichment of GO-slim terms in 200-gene sliding windows of increasing Spearman correlations was analyzed by two-tailed Fisher's exact test. Shown are GO-slim terms related to central carbon metabolism (CCM), and amino acid (AA) and protein metabolism, with at least one sliding-window with $p_{Fisher}<0.05$, indicating that genes with good protein–mRNA correlations are enriched in AA and protein metabolism, while genes with poor protein–mRNA correlations are enriched in CCM. (C) Proportion of amino acids in genes with different protein–transcript correlations were analyzed in 200-gene brackets of increasing ρ. Ten amino acids with the largest change with respect to ρ are shown. For each amino acid, log2 of the amino acid proportion of each 200-gene bracket, divided by the mean amino acid proportion of all genes, is shown. (D) Codon usage in genes with different protein–transcript correlations were analyzed in 200-gene brackets of increasing ρ. Codons of selected amino acids with the largest changes in codon usage are shown. For each codon, log2 of the codon usage of each 200-gene bracket, divided by the mean codon usage of all genes, is shown. Purines (A and G) are highlighted in gray.

The online version of this article includes the following figure supplement(s) for figure 7:

**Figure supplement 1.** Spearman correlation of mRNA and protein relative-abundances (% of total mol/mol) for each gene is calculated using data from both the GR and the NM experiments combined, and the distribution is shown demonstrating overall poor correlation.

abundance being well correlated only in a subset of genes (*Figure 8*, blue dashed arrow). Here, transcript expression is limited by purine availability, which in turn is constrained by intracellular Gly and Ser abundances; while protein expression may be controlled by nitrogen availability centered around Glu, Gln, Asp, and Asn concentrations in the cell (*Figure 8*, blue text and arrows). The protein abundance of genes with poor protein–transcript correlations, again enriched in CCM enzymes, are especially affected by this control mechanism (*Figure 8*, note *b*) because of the high proportion of these specific amino acids in their sequences.

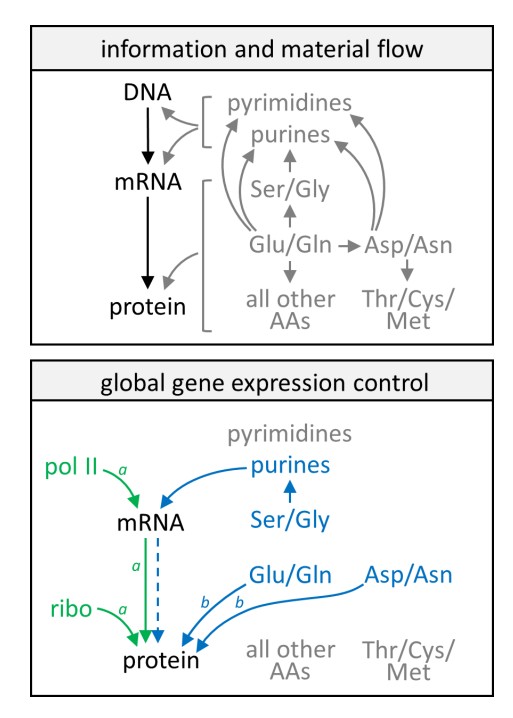

**Figure 8.** Model of gene expression control by growth rate and metabolism. In the upper panel, information flow in the central dogma (black text and arrows) and a simplified schematic of the flow of nitrogenous material (gray text and arrows) are given as reference points. In the lower panel, green text and arrows reflect gene expression control by the growth rate, showing that protein and mRNA expression track well with each other and each follows closely with levels of RNA polymerase II and ribosomes in the cell. Blue text and solid arrows reflect gene expression control by specific amino acids and nucleotides, and the blue dashed arrow represents that only in a subset of genes are protein abundance well correlated with transcript abundance when gene expression is subject to control by nitrogen metabolism. A more detailed description of the model can be found in the main text. AA, amino acids; pol II, RNA polymerase II; ribo, ribosome. [Note a]CCM genes are not subject to these controls by the cell growth rate. [Note b]CCM genes are particularly sensitive to these controls by nitrogen metabolism.

# Confounding effects of growth rate and metabolism in gene expression can be accounted for in gene expression profiling analyses

With an understanding of the effects of the cell growth rate and nitrogen metabolism on gene expression, these effects can be factored into gene expression analyses to allow the extraction of gene expression programs related specifically to the perturbation of interest. Here we showcase two such applications of our findings, first using a dataset in a distantly related yeast, *S. pombe*, and second using a dataset from a human cancer cell model.

*Marguerat et al., 2012* have shown that in *S. pombe*, gene expression in quiescent cells was characterized by a drastic downregulation of the total transcriptome compared to proliferating cells, with >85.3% transcripts being lowered by >2-fold. However, there is an inherent difference in growth rate between proliferating and quiescent cells, and quiescence was induced experimentally by nitrogen starvation (*Marguerat et al., 2012*). Together, we expect these differences in growth rate and nitrogen availability to influence the abundance of >90% of transcripts. What, then, are the gene expression changes induced by quiescence per se, without the confounding factors of growth rate and nitrogen limitation? To answer this question, we took the transcriptome abundance and protein translation rates of proliferating *S. pombe* and calculated the theoretical transcriptome abundance and protein translation rates if reduced growth rate and limitation of the nitrogen source ($NH_4$) were the only regulatory signals controlling differential gene expression. Comparisons between the measured and calculated transcriptome abundance (*Figure 9A*) and protein translation rates (*Figure 9B*) can then inform us of how gene expression is regulated by quiescence per se. As the expression of CCM genes generally do not follow the gene expression controls by growth rate and nitrogen metabolism, in the calculations of the theoretical transcriptome abundance and protein translation rates, we therefore considered them to be at a constant level for these genes. In effect this means that, for CCM genes (identified by the PomBase GO-slim term 'carbohydrate metabolic process') (*Lock et al., 2019*), the comparisons made in *Figure 9A,B* are equal to directly comparing the transcriptome abundance and protein translation rates between proliferating cells and quiescent cells. Our results showed that for most transcripts, the difference in abundance between proliferating and quiescent cells was largely in line with the difference in growth rate (*Figure 9A*). Quiescence per se induced a specific gene expression program by upregulating 707 transcripts (14%) and downregulating 574 transcripts (11%). In contrast, protein translation dynamics was extensively remodeled by quiescence, with 90% of $k_{sP}$ being upregulated (*Figure 9B*), showing that protein translation plays a surprisingly large role

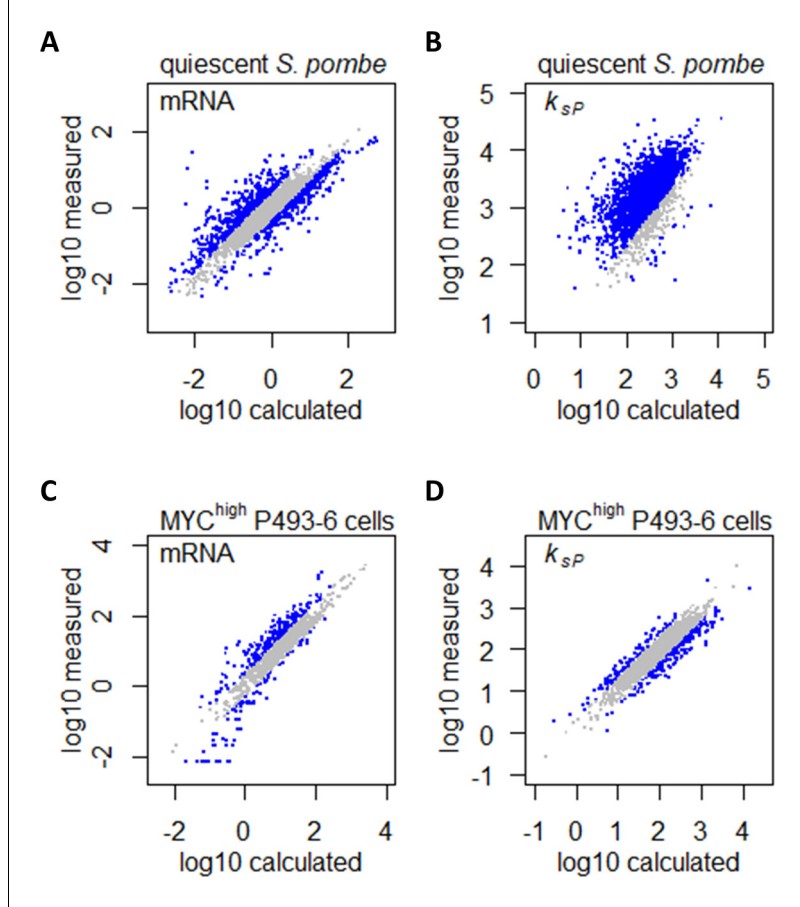

**Figure 9.** Confounding effects of growth rate and metabolism on gene expression can be accounted for in gene expression profiling analyses. (A) Measured mRNA abundances in quiescent *S. pombe* are compared to the calculated mRNA abundance, given the mRNA abundance in proliferating *S. pombe* and the difference in cell growth rate and nitrogen availability. Gray data points are those where calculated and measured values agree within 2-fold change ($-1 < \log_2 < 1$). (B) Protein translation rate $k_{sP}$ calculated in quiescent *S. pombe* cells are compared to the protein translation rate $k_{sP}$ calculated based on protein and mRNA abundance in proliferating *S. pombe* and the difference in cell growth rate and nitrogen limitation. Gray data points are those where calculated and measured values agree within 2-fold change ($-1 < \log_2 < 1$). (C) Measured mRNA abundance in MYC-overexpressing (MYC[high]) P493–6 cells are compared to the calculated mRNA abundance, given the mRNA abundance in P493–6 cells without MYC overexpression (MYC[low]) and the difference in cell growth rate and Gln usage. Gray data points are those where calculated and measured values agree within 2-fold change ($-1 < \log_2 < 1$). (D) Protein translation rate $k_{sP}$ calculated in MYC-overexpressing (MYC[high]) P493–6 cells are compared to the protein translation rate $k_{sP}$ calculated based on protein and mRNA abundance in P493–6 cells without MYC overexpression (MYC[low]) and the difference in cell growth rate and Gln usage. Gray data points are those where calculated and measured values agree within 2-fold change ($-1 < \log_2 < 1$).

The online version of this article includes the following figure supplement(s) for figure 9:

**Figure supplement 1.** Comparison between MYC-specific regulation of mRNA abundance and protein translation rate ($k_{sP}$).

in regulating gene expression in response to quiescence in *S. pombe*, which would be interesting to explore in future studies.

We then performed a similar re-analysis of gene expression profiling data in a human cancer cell model, to showcase that changes in gene expression caused by oncogenic perturbation can also be uncoupled from those that are induced by physiological parameters such as increased growth rate and altered metabolism. We mined gene expression datasets in the B-cell line P493–6 (*Lin et al., 2012*; *Feist et al., 2018*), which carries a conditional *c-Myc* allele that can be experimentally turned

on (MYC[high]), mimicking overexpression of the MYC transcription factor that is the driving oncogenic event in Burkitt's lymphoma (*Schuhmacher et al., 1999*). *Lin et al., 2012* quantified the absolute abundance of 1263 transcripts in the P493–6 cell system and found that 707 transcripts (56%) were upregulated by >2-fold in MYC[high] cells. Similarly, *Feist et al., 2018* quantified the absolute abundance of 1662 proteins and found that 871 proteins (52%) were upregulated by >2-fold in MYC[high] P493–6 cells. However, MYC overexpression in P493–6 also leads to faster cell growth (*Schuhmacher et al., 1999*) and Gln dependency (*Feist et al., 2018*), both of which could influence gene expression. We therefore sought to uncouple the effects of these physiological differences from the specific effects of MYC overexpression, which would help to elucidate direct MYC targets. Taking a similar approach as we have done in the re-analysis of the *S. pombe* dataset, we took the transcriptome abundance and protein translation rates of MYC[low] P493–6 cells and calculated the theoretical transcriptome abundance and protein translation rates if increased growth rate and usage of Gln as the nitrogen sources were the only regulatory signals controlling differential gene expression. Comparisons between the measured and calculated transcriptome abundance (*Figure 9C*) and protein translation rates (*Figure 9D*) can then inform us of how gene expression is regulated by the MYC oncogene, without the secondary effects of increased growth rate and Gln usage. As before, we also considered all CCM genes (identified by AmiGO 2 term 'cellular carbohydrate metabolic process') (*AmiGO Hub et al., 2009*) to be expressed constantly; thus, for CCM genes, the comparisons made in *Figure 9C,D* are equal to directly comparing the transcriptome abundance and protein translation rates between MYC[high] and MYC[low] P493–6 cells. Results indicated that the majority of the upregulated transcripts could be accounted for by changes in the cell growth rate and Gln usage (*Figure 9C*), while MYC overexpression specifically upregulated 309 transcripts (24%) and downregulated 167 transcripts (13%). Similarly, protein translation dynamics in MYC[high] P493–6 cells were also largely accounted for by changes in the cell growth rate and Gln usage (*Figure 9D*), with MYC overexpression causing the up- and downregulation of $k_{sP}$ for 101 (6%) and 212 (13%) proteins, respectively. Genes with upregulated mRNA abundance generally had lowered $k_{sP}$, and vice versa (*Figure 9—figure supplement 1*), indicating that protein translation plays a buffering role in MYC-overexpressing cancer cells, producing dampened effects on protein abundance compared to changes in transcript abundance, which will be an important consideration in efforts to develop MYC-targeting therapeutics.

## Discussion

Gene expression regulation is coupled with the physiological state of the cell such as growth rate and metabolism, which controls the availabilities of building blocks and energy, the abundance of RNA polymerases and ribosomes (*Klumpp et al., 2009*), as well as the sizes of transcriptomic and proteomic reserves (*Yu et al., 2020a*; *Mori et al., 2017*). Here we quantified the effects of cell growth rate and nitrogen metabolism on gene expression, by analyzing an orthogonal multi-omics dataset consisting of paired absolute-quantitative transcript–protein abundances of 22 steady-state yeast cultures. Overall, >90% of genes are regulated by the growth rate and/or nitrogen metabolism at both the transcript and protein levels, suggesting widespread confounding effects in gene expression profiling studies in which these factors are commonly neglected. The availabilities of RNA polymerase II and ribosomes are major contributors of differential gene expression with respect to the cell growth rate, while differential gene expression with respect to nitrogen metabolism occurs independently of these transcript and protein synthetic machineries. Additionally, transcription and translation are regulated in a coordinated manner in response to changing growth rate, but are regulated in distinct ways with respect to nitrogen metabolism, highlighting that interaction between different physiological parameters can be quite complex in regulating gene expression.

Previously we have shown that the cell growth rate determines the allocation of the proteome and transcriptome to different processes, including the relative abundance of RNA polymerases and ribosomes (*Yu et al., 2020a*; *Björkeroth et al., 2020*). Our results here indicate that these growth rate-dependent changes in the transcription and translation machineries directly influence the absolute abundance of ~90% of genes. However, the expression of genes involved in CCM are independent of the cell growth rate. Of note, this means that the allocation of the transcriptome and proteome (i.e. the relative abundance normalized to total transcripts and protein abundance, respectively) to CCM would decrease with growth rate (*Peebo et al., 2015*; *Valgepea et al., 2013*),

highlighting this important distinction between absolute and relative quantification of gene expression. Indeed, *Brauer et al., 2008* have shown using relative-quantitative data that carbon metabolic processes are enriched in genes that are negatively correlated with growth rate (*Brauer et al., 2008*). Here we showed that, in absolute-quantitative terms, CCM genes were expressed at constant levels independent of the cell growth rate. However, with 90% of genes having a positive relationship with the cell growth rate, in relative-quantitative terms, CCM genes would appear to be negatively regulated by the cell growth rate, consistent with the previous report (*Brauer et al., 2008*).

We also showed that nitrogen metabolism acts as a regulator of gene expression by consistently modulating the overall abundance of 96% of transcripts. Differential gene expression with respect to nitrogen metabolism is not dependent on how much RNA polymerases and ribosomes are present in the cell, but instead is closely associated with the availability of amino acids and nucleotides, as well as the proportion of specific amino acids in the protein sequence and codon usage in the transcript. Genes demonstrating the strongest uncoupling of transcription and translation as two distinct points of regulation – that is, genes whose transcript and protein abundances were correlated especially poorly – were enriched for CCM processes, consistent with previous work showing that this central metabolic pathway is regulated by complex mechanisms to ensure robust output (*Yu et al., 2020a*). These genes also contained higher proportions of Glu, Gln, Asp, and Asn, which are central to amino acid metabolism, and have higher usage of codons with higher purine content, together suggesting that amino acids and nucleotide metabolism may independently influence protein and transcript abundances, thereby contributing to a poor transcript–protein correlation.

In many experimental systems (e.g. drug treatment, wild type vs mutant, expression of transgenes), the physiological state of the cell is different between experimental conditions but is difficult to control for. Recent approaches to gain such control include the development of isogrowth gene expression profiling, whereby cell cultures treated with a 2D-drug gradient are sampled along a line of constant growth (the isobole), in order to dissect the gene expression changes without the confounding factor of changing growth rate (*Lukačišin and Bollenbach, 2019*). This methodology is however technically challenging and difficult to scale up. We show here that effects of growth rate and metabolism can be disentangled based on the results presented in this study. Of note, it is particularly important to take these effects into consideration when the fold-change in gene expression is small (i.e. a few fold), as very large changes (e.g. by orders of magnitude) are unlikely to be regulated by physiological parameters of the cell (*Klumpp et al., 2009*). One application of this is the gene expression changes in human cancer cells induced by overexpression of the oncogene MYC, which has been suggested to play an amplifier role to drive small increases in all expressed transcripts (*Lin et al., 2012*), but this effect is debated (*Walz et al., 2013*). Our analysis indicates that the increase in gene expression is in line with altered cell physiology during MYC-driven oncogenic transformation (*Feist et al., 2018*; *Schuhmacher et al., 1999*), thereby allowing MYC-specific effects of transcript expression and protein translation to be identified. Thus, by understanding the effects of gene expression exerted by the physiological state of the cell, our results allow better analyses of gene expression profiles to gain novel biological insights, a key conceptual goal of systems biology.

# Materials and methods

**Key resources table**

| Reagent type (species) or resource | Designation | Source or reference | Identifiers | Additional information |
|---|---|---|---|---|
| Strain, strain background (*Saccharomyces cerevisiae*) | CEN.PK113-7D | Nielsen lab | | MATa, MAL2-8c, *SUC2* |
| Commercial assay or kit | Qiagen RNeasy Mini Kit | Qiagen | Cat # 74106 | |
| Commercial assay or kit | Qubit RNA HS Assay Kit | Thermo Fisher | Cat # Q32852 | |

*Continued on next page*

*Continued*

| Reagent type (species) or resource | Designation | Source or reference | Identifiers | Additional information |
|---|---|---|---|---|
| Commercial assay or kit | Illumina TruSeq Stranded mRNA Library Prep Kit | Illumina | Cat # RS-122–2101 | |
| Commercial assay or kit | Pierce BCA Protein Assay Kit | Thermo Fisher | Cat # 23225 | |
| Commercial assay or kit | TMT10plex Isobaric Label Reagent Set | Thermo Fisher | Cat # 90110 | |
| Commercial assay or kit | SCIEX aTRAQ Reagents Application Kit | Sciex | Cat # 4442678 | |
| Peptide, recombinant protein | UPS2 Proteomics Dynamic Range Standard | Sigma–Aldrich | Cat # UPS2-1SET | |
| Software, algorithm | Proteome Discoverer 2.2 | Thermo Fisher | | |
| Software, algorithm | Mascot 2.5.1 | Matrix Science | | |
| Software, algorithm | R 4.0.3 | R Project | | |

## Culture conditions

The yeast *S. cerevisiae* CEN.PK113-7D (MATa, MAL2-8c, *SUC2*) was used for all experiments. Cells were stored in aliquoted glycerol stocks at −80°C. Chemostat experiments were carried out in DAS-GIP 1L bioreactors (Jülich, Germany) equipped with off-gas analysis, pH, temperature, and dissolved oxygen sensors. Chemostat experiments were carried out at 30°C, pH 5, working volume 0.5 L, aeration 1 vvm, $pO_2 > 30\%$, and agitation speed 800 rpm. The glucose and nitrogen source concentrations are in *Supplementary file 1a*. Additional media components are as follows: $KH_2PO_4$, 3 g $L^{-1}$; $MgSO_4 \cdot 7H_2O$, 0.5 g $L^{-1}$; trace metals solution, 1 mL $L^{-1}$; vitamin solution, 1 mL $L^{-1}$; and antifoam, 0.1 mL $L^{-1}$. The trace metal solution contained: EDTA (sodium salt), 15.0 g $L^{-1}$; $ZnSO_4 \cdot 7H_2O$, 4.5 g $L^{-1}$; $MnCl_2 \cdot 2H_2O$, 0.84 g $L^{-1}$; $CoCl_2 \cdot 6H_2O$, 0.3 g $L^{-1}$; $CuSO_4 \cdot 5H_2O$, 0.3 g $L^{-1}$; $Na_2MoO_4 \cdot 2H_2O$, 0.4 g $L^{-1}$; $CaCl_2 \cdot 2H_2O$, 4.5 g $L^{-1}$; $FeSO_4 \cdot 7H_2O$, 3.0 g $L^{-1}$; $H_3BO_3$, 1.0 g $L^{-1}$; and KI, 0.10 g $L^{-1}$. The vitamin solution contained: biotin, 0.05 g $L^{-1}$; p-amino benzoic acid, 0.2 g $L^{-1}$; nicotinic acid, 1 g $L^{-1}$; Ca-pantothenate, 1 g $L^{-1}$; pyridoxine-HCl, 1 g $L^{-1}$; thiamine-HCl, 1 g $L^{-1}$; and myo-inositol, 25 g $L^{-1}$. The measured nutrient fluxes in all chemostats are given in *Supplementary file 1e*.

## Sampling from bioreactor

The dead volume was collected with a syringe and discarded. For transcriptome sampling, biomass was collected from the reactor with a syringe and injected into chilled 50 mL Falcon tubes filled with 35 mL crushed ice. Samples were centrifuged for 4 min at 3000 × g at 4°C; cell pellets were washed once with 1 mL of chilled water, transferred into Eppendorf tubes, flash frozen in liquid nitrogen, and stored at −80°C until analysis. For proteome sampling, biomass was collected from the reactor with a syringe and injected into 50 mL Falcon tubes chilled on ice. Samples were centrifuged for 4 min at 3000 × g at 4°C; cell pellets were washed once with 20 mL of chilled $dH_2O$, washed again with 1 mL of chilled water, transferred into Eppendorf tubes, flash frozen in liquid nitrogen, and stored at −80°C until analysis. For intracellular amino acid measurements, biomass was collected from the reactor with a syringe and injected into −80°C MetOH at 10 times the volume of the sample. Samples were centrifuged to 4 min at 3000 × g at −20°C, and pellets were flash frozen in liquid nitrogen and stored at −80°C until analysis. Biomass determination was done by filtration of the culture broth on pre-weighed filter paper, drying in a microwave at 360 W for 20 min, and desiccating in a desiccator for >3 days.

## RNA sequencing

RNA was extracted using Qiagen RNeasy Mini Kit (Qiagen, Hilden, Germany) according to manufacturer's protocol. RNA integrity was examined using a 2100 Bioanalyzer (Agilent Technologies, Santa Clara, CA). RNA concentration was determined using a Qubit RNA HS Assay Kit (Thermo Fisher, Waltham, MA). The Illumina TruSeq Stranded mRNA Library Prep Kit (Illumina, San Diego, CA) was used to prepare mRNA samples for sequencing. Paired-end sequencing (MID Output 2 × 75 bp)

was performed on an Illumina NextSeq 500 (Illumina, San Diego, CA). Reads were quality controlled, mapped to the *S. cerevisiae* reference genome (Ensembl R64-1-1), and counted using the nf-core RNAseq pipeline (SciLifeLab, Stockholm, Sweden), available at https://nf-co.re/rnaseq.

## Sample preparation for proteomic analysis

Liquid chromatography–mass spectrometry (LC–MS)-based proteomic analysis was performed largely as described in our previous publication (*Yu et al., 2020a*). Yeast cell pellets were suspended in the lysis buffer (2% sodium dodecyl sulfate, 50 mM triethylammonium bicarbonate [TEAB]) and homogenized using a FastPrep−24 instrument (Matrix C, 1 mm silica spheres; MP Biomedicals, Akron, OH) for 5 repeated 40 s cycles at 6.5 m/s, with 30–60 s breaks in-between. The lysate was centrifuged at 21,100 x g for 20 min, the supernatant was transferred to new tubes and diluted five times with the lysis buffer, and protein concentration was determined using Pierce BCA Protein Assay Kit (Thermo Fischer Scientific, Waltham, MA) and the Benchmark Plus microplate reader (Bio-Rad Laboratories, Hercules, CA), with bovine serum albumin solutions as standards. Pooled reference sample ('Reference') has been prepared by mixing equal amounts (according to BCA (bicinchoninic acid) measurement) from each supernatant.

For the TMT-based relative quantification, aliquots containing 25 μg of protein were taken from each experimental sample and from the pooled reference sample; an aliquot of 50 μg of the Reference sample was spiked with 10.6 μg of the UPS2 Proteomics Dynamic Range Standard (Sigma–Aldrich, Saint-Louis, MO) for the IBAQ (*Schwanhäusser et al., 2011*) quantification. Each sample was reduced by the addition of 2 M DL-dithiothreitol to a final concentration of 100 mM and incubated at 56°C for 30 min. The samples were processed according to the modified filter-aided sample preparation method (*Wiśniewski et al., 2009*). In short, reduced samples were diluted to 300 μL by addition of 8 M urea, transferred onto Nanosep 30 k Omega filters (Pall Corporation, Port Washington, NY), and washed two times with 200 μL of 8 M urea. The free cysteine side chains were alkylated with 10 mM methyl methanethiosulfonate solution in digestion buffer (1% sodium deoxycholate [SDC], 50 mM TEAB) for 30 min at room temperature, and the filters were then repeatedly washed with digestion buffer. Trypsin in digestion buffer was added (250 ng for TMT samples/500 ng for the IBAQ sample), and the sample was incubated at 37°C for 3 hr, then another aliquot of trypsin (250/500 ng) was added and incubated overnight. Digested peptides were collected by centrifugation at 9500 × g for 20 min, followed by a wash with 20 μL of the digestion buffer and centrifugation at 9500 × g for 20 min. Peptide samples for relative quantification were labeled using the five sets TMT 10plex isobaric reagents according to the manufacturer's instructions (Thermo Scientific), combined into five pooled TMT samples, concentrated using vacuum centrifugation, and SDC was removed by acidification with 10% TFA (Trifluoroacetic acid) and subsequent centrifugation at 16,000 × g for 10 min. The digested sample for IBAQ label-free quantification was acidified with 10% TFA and centrifuged at 16,000 × g for 10 min to remove the precipitated deoxycholic acid.

The TMT sets were fractionated into 44 primary fractions by basic reversed-phase chromatography using a Dionex Ultimate 3000 UPLC system (Thermo Fischer Scientific, Waltham, MA). Peptide separations were performed on a reversed-phase XBridge BEH C18 column (3.5 μm, 3.0 × 150 mm, Waters Corporation) and a linear gradient from 3% to 40% solvent B over 17 min followed by an increase to 100% B over 5 min. Solvent A was 10 mM aqueous ammonium formate at pH 10.0, and solvent B was 90% acetonitrile, 10% 10 mM ammonium formate at pH 10.00. The primary fractions were concatenated into final 20 fractions (1+21+41, 2+22+42, . . . 20+40), evaporated, and reconstituted in 15 μL of 3% acetonitrile, 0.2% formic acid for nLC–MS analysis.

The pooled sample for the label-free quantification was fractionated into 44 primary fractions that were concatenated into 10 final fractions (1+11+21+31+41, 2+12+22+32+42, . . . 10+20+30+40). The final fractions were evaporated and reconstituted in 15 μL of 3% acetonitrile, 0.2% formic acid for nLC–MS analysis.

## LC–MS for proteomic analysis

All samples were analyzed on an Orbitrap Fusion Tribrid mass spectrometer interfaced with Easy-nLC1200 liquid chromatography system (both Thermo Fisher Scientific). Peptides were trapped on an Acclaim Pepmap 100 C18 trap column (100 μm × 2 cm, particle size 5 μm, Thermo Fischer Scientific) and separated on an in-house packed analytical column (75 μm × 30 cm, particle size 3 μm,

Reprosil-Pur C18, Dr. Maisch) using the linear gradients with 0.2% formic acid in water as a solvent A and 80% acetonitrile, 0.2% formic acid as solvent B with the flow of 300 nL/min.

Each TMT-labeled fraction was separated on a 90 min gradient from 5% to 35% B over 75 min, from 35% to 100% B over 5 min, and 100% B for 10 min. Each fraction for the label-free IBAQ quantification was injected three times and analyzed using the same elution gradient.

For the TMT-based relative quantification, MS scans were performed at 120,000 resolution, m/z range 380–1380 with the wide quadrupole isolation and AGC target 4e5; the most abundant precursors with charges 2–7 were selected for fragmentation over the 3 s cycle time with the dynamic exclusion duration of 60 s. Precursors were isolated with a 0.7 Da window, fragmented by collision-induced dissociation (CID) at 35% collision energy with a maximum injection time of 50 ms and AGC target 1e4, and the $MS^2$ spectra were detected in the ion trap followed by the synchronous isolation of the 10 most abundant $MS^2$ fragment ions within m/z range of 400–1400 and fragmentation by higher-energy collision dissociation at 65%; the resulting $MS^3$ spectra were detected in the Orbitrap at 50,000 resolution with m/z range 100–500, maximum injection time 105 ms and AGC target 1e5.

For label-free quantification IBAQ experiment, MS scans were performed at 120,000 resolution, m/z range 380–1380 with the wide quadrupole isolation and AGC target 2e5; the most abundant precursors with charges 2–7 were selected for fragmentation over the 1 s cycle time with the dynamic exclusion duration of 45 s. Precursors were isolated with a 1.2 Da window, fragmented by CID at 35% collision energy with a maximum injection time of 50 ms and AGC target 1e4, and the $MS^2$ spectra were detected in the ion trap.

## Proteomic raw data processing

Peptide and protein identification and quantification was performed using Proteome Discoverer version 2.2 (Thermo Fisher Scientific) with Mascot 2.5.1 (Matrix Science, London, UK) as a database search engine. The Baker's yeast (*S. cerevisiae* ATCC 204508/S288c) reference proteome database was downloaded from Uniprot (February 2018, 6049 sequences) and used for the database search on the TMT-based relative quantification files; the concatenated database containing the yeast sequences and the 48 UPS protein sequences was used for the processing of the UPS2-spiked files.

Precursor masses have been re-calibrated with the initial mass tolerance of 20 ppm prior to the main database search. For the TMT relative quantification data and the label-free IBAQ data, trypsin with one missed cleavage was used as a cleavage rule, MS peptide tolerance was set to 5 ppm, and $MS^2$ tolerance for identification was set to 600 milli mass units (mmu). Variable modifications of methionine oxidation and fixed modifications of cysteine methylthiolation were used for both sub-data sets. TMT-6 label on lysine and peptide N-termini was set as a fixed modification for the TMT data. Percolator was used for the peptide-spectrum match validation with the strict FDR threshold of 1%.

Precursor ion quantification was accomplished via the Minora feature detection node in Proteome Discoverer 2.2, with the maximum peak intensity values used for quantification. Abundance values of all unique and shared peptides were used for the IBAQ calculation. Abundances from the three technical replicates were averaged and divided by the number of theoretically observable peptides for a protein to yield the IBAQ intensity (the number of observable peptides being calculated using an in-house Python script). The known absolute amount values of the UPS2 standard proteins were used to scale the log-transformed IBAQ intensity values versus log-transformed protein concentration. Only UPS2 proteins with at least two identified unique peptides were used for scaling.

The TMT reporter ions were identified in the $MS^3$ HCD spectra with a mass tolerance of 3 mmu, and the signal-to-noise (S/N) abundances of the reporter ions of all unique and shared peptides were used for relative quantification with the minimal average reporter S/N threshold set at 19 and the co-isolation threshold at 100. The resulting reporter abundance values for each sample were normalized within Proteome Discoverer 2.2 on the total peptide amount.

## Intracellular amino acid measurements

To extract intracellular amino acid, 1–2 mL of boiling 75% ethanol was poured directly onto the cell pellets. Samples were vortexed for 1 min, boiled for 3 min, and placed on ice until cool. Samples were centrifuged for 15 min at 13,000 × g at 4°C. Supernatants containing intracellular amino acids were collected and stored at −20°C until labeling and analysis. Amino acids were labeled using the

SCIEX aTRAQ Reagents Application Kit (Danaher, Washington, DC) with aTRAQ Reagent Δ8 and mixed with internal standards pre-labeled with aTRAQ Reagent Δ0, according to manufacturer's protocol. Samples were analyzed using a SCIEX QTRAP 6500+ system (Danaher, Washington, DC) with a Nexera UHPLC system (Shimadzu, Japan), on a SCIEX AAA column (150 × 4.6 mm) at an oven temperature of 50℃. Mobile phase A contained 0.1% formic acid and 0.01% heptafluorobutyric acid in water; mobile phase B contained 0.1% formic acid and 0.01% heptafluorobutyric acid in methanol. The flow rate was 0.8 mL min$^{-1}$, with a gradient profile was as follows: 0 min, 2% B; 6 min, 40% B; 10 min, 40% B; 11 min, 90% B; 12 min, 90% B; 13 min, 2% B; 18 min, 2% B. The retention time, precursor (Q1 m/z), and product (Q3 m/z) for each amino acid and internal standard were as described by the manufacturer's protocol. The mass spectrometer was set to monitor the transitions with the following ion source parameters: CUR (curtain gas) 30, CAD (collision activated dissociation) MED, IS (ionization voltage) 5500, TEM (temperature) 500, GS1 (source gas 1) 60 and GS2 (source gas 2) 50, and with compound parameters: DP (declustering potential) 30, EP (entrance potential) 10, CE (collision energy) 30, and CXP (collision cell exit potential) 5.

## Data processing and analysis

For transcriptomics, the absolute concentrations of 31 transcripts with >10 FPKM, and covering the entire dynamic expression range, were measured using lysates of *S. cerevisiae* CEN.PK 113-7D cells from the J. Nielsen lab (Chalmers University of Technology, Göteborg, Sweden). Linear regression between the absolute concentrations of these mRNAs and their corresponding FPKM values from RNAseq was performed to obtain the slope and y-intercept, which were used to quantify all mRNA in this study. The calculated mRNA abundance was then scaled to the total RNA content measured by Qubit RNA HS Assay Kit (Thermo Fisher, Waltham, MA). For proteomics, linear regression was calculated between the known log10 UPS2 concentrations and the log10 IBAQ abundances for the corresponding proteins. The resulting regression coefficients were used to estimate the absolute concentrations of all detected yeast proteins.

Protein translation rate $k_{sP}$ (*Lahtvee et al., 2017*; *Schwanhäusser et al., 2011*) for each protein $j$ was calculated by:

$$k_{sP,j} = \frac{C_{prot,j} \cdot \left( k_{dP,j} + \mu \right)}{C_{mRNA,j}}$$

The gene-specific protein degradation rate ($k_{dP}$) for *S. cerevisiae* was mined from *Lahtvee et al., 2017*. Protein degradation rate for *S. pombe* was mined from *Christiano et al., 2014*. Protein degradation rate for a cancer cell line (HeLa) was mined from *Boisvert et al., 2012*. Where the $k_{dP,j}$ was unavailable in the mined datasets, the median of all $k_{dP}$ in the dataset was used.

As transcript and protein abundance as well protein translation rates span several orders of magnitude, for ease of visualization we present most of our analyses using scaled expression levels. In the GR dataset, the scaling is done as follows: for each gene, linear regression is performed between the data being scaled (transcript abundance, protein abundance, protein translation rate, or amino acid abundance) and the growth rate. The y-intercept is obtained, and the data being scaled is divided by the y-intercept. This scaled data is then plotted on a log2 scale, which can be interpreted intuitively as the fold-change from when the cells are growing at a theoretical growth rate of 0. In the NM dataset, the scaling is done as follows: for each gene, the data being scaled (transcript abundance, protein abundance, protein translation rate, or amino acid abundance) is divided by the mean of the data in nitrogen-limited Phe and Ile as nitrogen source conditions (samples 34, 35, 36, 40, 41, and 42, in *Supplementary file 1a*). This scaled data is then plotted on a log2 scale, and here the interpretation is the fold-change from when cells are growing with severe nitrogen restriction (i.e. nitrogen is limiting and the nitrogen source is non-preferred).

## Data and code availability

Processed quantitative transcriptomics and proteomics data are in *Supplementary files 1b,c*. Processed intracellular amino acid concentrations are in *Supplementary files 1d*. Raw RNAseq data are available at ArrayExpress, accession E-MTAB-9117 (*Yu et al., 2020b*). The mass spectrometry proteomics data are deposited to the Proteome Xchange Consortium via the PRIDE (*Perez-Riverol et al., 2019*) partner repository with dataset identifier PXD021218 (*Vorontsov and Nielsen, 2020*). GO-

slim terms for *S. cerevisiae* genes are available from the Saccharomyces Genome Database (*Cherry et al., 2012*; *The Gene Ontology Consortium, 2019*). GO-slim terms for *S. pombe* genes are available from PomBase (*Lock et al., 2019*; *The Gene Ontology Consortium, 2019*). GO terms for human genes are available from AmiGO 2 (*AmiGO Hub et al., 2009*; *The Gene Ontology Consortium, 2019*). Custom code are available in the GitHub repository https://github.com/SysBio-Chalmers/YeastAbsQuantForCCMRegulation (*Yu, 2021*).

## Acknowledgements

We thank the National Bioinformatics Infrastructure Sweden (NBIS) for discussions. We thank Pannipa Pornpitakpong and Alexandra Hoffmeyer (Technical University of Denmark) for conducting RNA sequencing. The Proteomics Core Facility of the University of Gothenburg is grateful to the Inga-Britt and Arne Lundbergs Forskningsstiftlese for the donation of the Orbitrap Fusion Tribrid MS instrument. This research was supported by funding from the Novo Nordisk Foundation (grant number NNF10CC1016517) and the Knut and Alice Wallenberg Foundation.

## Additional information

### Competing interests
Jens Nielsen: is the CEO of the BioInnovation Institute, Denmark. The other authors declare that no competing interests exist.

### Funding

| Funder | Grant reference number | Author |
|---|---|---|
| Novo Nordisk Fonden | NNF10CC1016517 | Rosemary Yu<br>Jens Nielsen |
| Knut och Alice Wallenbergs Stiftelse | | Rosemary Yu<br>Jens Nielsen |

The funders had no role in study design, data collection and interpretation, or the decision to submit the work for publication.

### Author contributions

Rosemary Yu, Conceptualization, Data curation, Formal analysis, Methodology, Writing - original draft; Egor Vorontsov, Data curation, Formal analysis, Methodology, Writing - review and editing; Carina Sihlbom, Supervision, Writing - review and editing; Jens Nielsen, Conceptualization, Supervision, Funding acquisition, Writing - review and editing

### Author ORCIDs

Rosemary Yu https://orcid.org/0000-0001-9901-4055
Jens Nielsen https://orcid.org/0000-0002-9955-6003

### Decision letter and Author response
Decision letter https://doi.org/10.7554/eLife.65722.sa1
Author response https://doi.org/10.7554/eLife.65722.sa2

## Additional files

### Supplementary files
• Supplementary file 1. Containing all data. (**a**) Chemostat culture conditions and sample ID. (**b**) Absolute protein and mRNA abundances (fmol/mgDW) in yeast chemostat cultures (see a for culture conditions and sample ID); cluster information; and protein–mRNA correlations (Pearson and Spearman). (**c**) Absolute protein and mRNA abundances (fmol/mgDW) mined from Xia et al. and cluster information. (**d**) Intracellular amino acid (AA) abundance (μmol/gDW) in yeast chemostat cultures

(see a for culture conditions and sample ID). (**e**) Measured metabolic fluxes in yeast chemostat cultures (see a for culture conditions and sample ID).

- Transparent reporting form

## Data availability

Processed quantitative transcriptomics and proteomics data are in Supplementary file 1b and c. Processed intracellular amino acid concentrations are in Supplementary file 1d. Raw RNAseq data are available at ArrayExpress, accession E-MTAB-9117. The mass spectrometry proteomics data are deposited to the Proteome Xchange Consortium via the PRIDE partner repository with dataset identifier PXD021218.

The following datasets were generated:

| Author(s) | Year | Dataset title | Dataset URL | Database and Identifier |
|---|---|---|---|---|
| Yu R, Vorontsov E, Sihlbom C, Nielsen J | 2021 | Quantification of global gene expression control by growth rate and metabolic cues | http://www.ebi.ac.uk/arrayexpress/experiments/E-MTAB-9117 | ArrayExpress, E-MTAB-9117 |
| Vorontsov E, Nielsen J | 2021 | Quantifying gene expression control by growth and metabolism reveals distinct regulation of central carbon metabolism genes in yeast | https://www.ebi.ac.uk/pride/archive/projects/PXD021218/ | PRIDE, PXD021218 |

The following previously published datasets were used:

| Author(s) | Year | Dataset title | Dataset URL | Database and Identifier |
|---|---|---|---|---|
| Xia J, Sanchez BJ, Chen Y, Campbell K, Kasvandik S, Nielsen J | 2019 | Tanscriptome studies of yeast cell under a series of specific growth rates | https://www.ncbi.nlm.nih.gov/bioproject/?term=PRJNA523289 | NCBI BioProject, PRJNA523289 |
| Xia J, Sanchez BJ, Chen Y, Campbell K, Kasvandik S, Nielsen J | 2019 | Proteome studies of yeast cell under a series of specific growth rates | https://www.ebi.ac.uk/pride/archive/projects/PXD012891/ | PRIDE, PXD012891 |

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
