## [Decision Letter]

**Acceptance summary:**

This study has generated a large amount of solid data in the form of a new multi-ome database containing combinations of absolute mRNA quantities, proteome and amino acid concentrations in a set yeast population grown in various conditions in chemostats. Apart from being a valuable resource for colleagues, analysis of the data confirms the results of several previous seminal studies, suggesting how protein levels correlate with mRNA abundance, growth parameters and/or amino acid and nucleotide levels.

**Decision letter after peer review:**

Thank you for submitting your article "Quantifying absolute gene expression profiles reveals distinct regulation of central carbon metabolism genes in yeast" for consideration by *eLife*. Your article has been reviewed by three peer reviewers, including Kevin J Verstrepen as the Reviewing Editor and Reviewer #1, and the evaluation has been overseen by Patricia Wittkopp as the Senior Editor.

Essential Revisions:

All reviewers agree that while no further experimental work is needed, the paper needs to be thoroughly revised before it can be considered for publication. The consensus is that in its current form, the text does not make it clear what the exact goals and setup of the experiments were. In addition, some terminology seems ill-defined.

We would therefore ask a thorough revision to better highlight the central goal and strategy of the study, and better explain the logic behind the conclusions.

The individual reports below contain more detail that will hopefully help as you rework the text. As you can see, reviewers 2 and 3 in particular share the feeling that the flow and logic of the paper is lacking. We are optimistic that a thorough re-write will make the potential impact of this work that we see more apparent to future readers, but if there is any doubt, we will send the work back to reviewers for additional input.

Reviewer #1 (Recommendations for the authors):

Remark

While it is tempting to make speculations based on the observed correlations, and while I do believe that in general, the suggestions that the authors make are likely correct, it remains important to point out that correlations do not imply direct causation. Therefore, I'd suggest pointing this out explicitly, for example in the section that concludes that "increasing growth rate modulated global gene expression via coordinated increases in both transcription and translation, by increasing the availability of RNA polymerase II and ribosomes." (and other similar parts).

Question

Is there any difference in the correlation between mRNA levels and protein abundance for proteins that contain different proportions of specific amino acids? And what about codon usage? One would expect that these are perhaps more important factors when different nitrogen sources are used, and less so when growth rate is controlled through the availability of a good nitrogen source? If so, this would perhaps provide further evidence for the hypotheses.

Reviewer #2 (Recommendations for the authors):

Key point: Through a major rewrite they authors could better explain the intent of the paper, and the significance of the manuscript overall.

Important pieces of information are not explained, or are left to the Materials and methods section or another paper. For instance, the calculation of the protein translation rate is left to the Materials and methods section (even though it seems to be an important point) and from the main text, it is not clear how the "coarse-graining approach" (the "framework"?) works, even though the "approach/framework" could be the key aspect of the paper (see above). Notably, Shahrezaei and Marguerat, 2015, which is the paper the authors refer to with regards to the "coarse-graining approach" is a Current Opinion paper, which has no Materials and methods section. Thus, I really don't understand what the authors mean with the "coarse-graining approach" and what they did.

Maybe due to this vagueness, due to the relatively limited explanations, or possibly also due to my insufficient understanding of the material, I had trouble getting the model that the authors present (in Figure 7), and the analyses that the authors perform in the following.

Reviewer #3 (Recommendations for the authors):

This paper would be really valuable if rethought and rewritten from the base up. Decide on the main points you want to make (this part seems to be mostly done), and then devote a section/paragraph to each of them containing the supporting evidence. This will help the reader tremendously. Right now, it just feels like a treasure trove of anecdotes that is complicated to read.

---

## [Author Response]

Essential Revisions:All reviewers agree that while no further experimental work is needed, the paper needs to be thoroughly revised before it can be considered for publication. The consensus is that in its current form, the text does not make it clear what the exact goals and setup of the experiments were. In addition, some terminology seems ill-defined.We would therefore ask a thorough revision to better highlight the central goal and strategy of the study, and better explain the logic behind the conclusions.The individual reports below contain more detail that will hopefully help as you rework the text. As you can see, reviewers 2 and 3 in particular share the feeling that the flow and logic of the paper is lacking. We are optimistic that a thorough re-write will make the potential impact of this work that we see more apparent to future readers, but if there is any doubt, we will send the work back to reviewers for additional input.

We have re-structured our manuscript and re-written many sections of the text to improve the flow and logic of the paper. We have added text stating the goals, experimental setup, and analysis strategy of the study, in the Introduction and Results sections where appropriate. Ill-defined terminology were either removed or given a clearer description in the text.

Reviewer #1 (Recommendations for the authors):RemarkWhile it is tempting to make speculations based on the observed correlations, and while I do believe that in general, the suggestions that the authors make are likely correct, it remains important to point out that correlations do not imply direct causation. Therefore, I'd suggest pointing this out explicitly, for example in the section that concludes that "increasing growth rate modulated global gene expression via coordinated increases in both transcription and translation, by increasing the availability of RNA polymerase II and ribosomes." (and other similar parts).

We have added this in the revised manuscript, in a newly added section of text which describes the goals and strategy of the study and also highlights that these correlations do not imply direct causation (Results section).

QuestionIs there any difference in the correlation between mRNA levels and protein abundance for proteins that contain different proportions of specific amino acids? And what about codon usage? One would expect that these are perhaps more important factors when different nitrogen sources are used, and less so when growth rate is controlled through the availability of a good nitrogen source? If so, this would perhaps provide further evidence for the hypotheses.

We have added this analysis in the revised manuscript (Figure 7C-D and Results section). Indeed this has shown that proportions of Gln, Glu, Asp, and Asn in the protein sequence (among others), and usage of codons with higher purine content, are different between genes with high vs. low protein-transcript correlations, supporting our model that amino acid and nucleotides can regulate gene expression independently of RNA polymerases and ribosomes.

Reviewer #2 (Recommendations for the authors):Key point: Through a major rewrite they authors could better explain the intent of the paper, and the significance of the manuscript overall.

We have re-structured our manuscript and re-written many sections of the text to increase clarity and flow. We have added text stating the goals, experimental setup, and analysis strategy of the study where appropriate.

Important pieces of information are not explained, or are left to the Materials and methods section or another paper. For instance, the calculation of the protein translation rate is left to the Materials and methods section (even though it seems to be an important point) and from the main text.

We have moved the calculations of the protein translation rate in the Results section and described it more clearly.

It is not clear how the "coarse-graining approach" (the "framework"?) works, even though the "approach/framework" could be the key aspect of the paper (see above). Notably, Shahrezaei and Marguerat, 2015, which is the paper the authors refer to with regards to the "coarse-graining approach" is a Current Opinion paper, which has no Materials and methods section. Thus, I really don't understand what the authors mean with the "coarse-graining approach" and what they did.

We apologize for the error in using this terminology and the incorrect reference. We have now removed the term “coarse-graining approach” and instead given a detailed description of the method used (Results section).

Maybe due to this vagueness, due to the relatively limited explanations, or possibly also due to my insufficient understanding of the material, I had trouble getting the model that the authors present (in Figure 7), and the analyses that the authors perform in the following.

We have re-structured the manuscript and a more detailed description of the model (now Figure 8) is given in the Results section and in the figure legend. We have also added a section of text which describes the goals and strategy of the study. Throughout the text we have added explanations to our findings to improve clarity.

Reviewer #3 (Recommendations for the authors):This paper would be really valuable if rethought and rewritten from the base up. Decide on the main points you want to make (this part seems to be mostly done), and then devote a section/paragraph to each of them containing the supporting evidence. This will help the reader tremendously. Right now, it just feels like a treasure trove of anecdotes that is complicated to read.

We have re-structured our manuscript and re-written many sections of the text to improve the flow and logic of the paper. We have added text stating the goals, experimental setup, and analysis strategy of the study, in the Introduction and Results sections where appropriate. The model is described in more detail in the text and in the legend.